# Amino Acid Requirements for Nile Tilapia: An Update

**DOI:** 10.3390/ani13050900

**Published:** 2023-03-01

**Authors:** Wilson Massamitu Furuya, Thais Pereira da Cruz, Delbert Monroe Gatlin

**Affiliations:** 1Department of Animal Science, State University of Ponta Grossa, Ponta Grossa 84030-900, Brazil; 2Animal Science Graduate Degree Program, State University of Maringá, Maringá 87020-900, Brazil; 3Department of Ecology and Conservation Biology, Texas A&M University System, College Station, TX 77840, USA

**Keywords:** amino acid requirement, ideal amino acid ratio, growth performance, health status, Nile tilapia

## Abstract

**Simple Summary:**

The concept of optimizing growth performance and supporting fish health through precise amino acid (AA) nutrition is well-accepted in current Nile tilapia, *Oreochromis niloticus*, farming. Emerging studies have evidenced the crucial role of essential amino acids (EAAs) and non-essential amino acids (NEAAs) beyond protein synthesis, regulating growth and reproductive performance, health status, fillet yield, and flesh quality responses of Nile tilapia. Balanced AAs can contribute to accurately implementing the “Precision Nutrition” concept and may help assess the economic dimension of the aquaculture system. Additionally, information on the precise dietary AA requirements may help produce environmentally sustainable diets for Nile tilapia farming in compliance with sustainability principles.

**Abstract:**

This review aims to consolidate the relevant published data exploring the amino acid (AA) requirements of Nile tilapia, *Oreochromis niloticus*, and to reach a new set of recommendations based on those data. There are still inconsistencies in lysine, sulfur-containing AA, threonine, tryptophan, branched-chain AA, and total aromatic AA recommendations in data that have appeared since 1988. This review finds that strain, size, basal diet composition, and assessment method may have contributed to the inconsistencies in AA recommendations. Currently, the expansion of precision AA nutrition diets for Nile tilapia is receiving more attention because of the demand for flexibility in widespread ingredient substitutions which will allow compliance with environmentally sustainable principles. Such approaches involve changes in diet ingredient composition with possible inclusions of non-bound essential and non-essential AAs. Increasing the inclusion of non-bound AAs into Nile tilapia diets may modify protein dynamics and influence AA requirements. Emerging evidence indicates that not only essential but also some non-essential amino acids regulate growth performance, fillet yield, and flesh quality, as well as reproductive performance, gut morphology, intestinal microbiota, and immune responses. Thus, this review considers current AA recommendations for Nile tilapia and proposes refinements that may better serve the needs of the tilapia industry.

## 1. Introduction

Global tilapia production is projected to continue growing until 2031 through the sustainable management and utilization of natural resource principles based on improved nutrition practices [1]. In this way, the growth performance, reproduction, and health of tilapias have been improved by genetic selection breeding programs coupled with precision nutritional strategies to meet this increasing demand. However, this need has generated challenges for the tilapia industry, including concerns of food security, food safety, feed ingredient shortages, diseases, and environmental issues.

Emerging evidence suggests a variety of economically and environmentally sustainable feedstuffs for use in aquafeeds [2,3,4,5]. One of these groups is the plant-protein feedstuffs; however, vegetable feedstuffs may contain antinutrients and limiting amounts of certain amino acids (AAs) that may impair protein synthesis [3,6,7,8,9]. Consistently, the deficiency of a single essential AA may impair several physiological functions and, subsequently, growth performance [10,11,12]. Emerging evidence suggests that AAs also act as signaling molecules regulating protein synthesis [13,14] and energy metabolism in the Nile tilapia, *Oreochromis niloticus* [15]. Therefore, the provision of tilapia feeds closely matching optimum AA requirements is a crucial strategy for overcoming various challenges, including optimizing sustainable raw materials, lowering feeding costs, and attenuating nitrogen loss into the environment.

Fish, like other animals, synthesize body proteins from AAs that are provided in the diet as well as some AAs that can be synthesized in the body from precursors. Those which must be provided in the diet due to lack of endogenous synthesis capabilities have traditionally been referred to as dietary indispensable or essential amino acids (EAAs). Classical fish nutrition textbooks consider arginine, histidine, isoleucine, leucine, lysine, methionine, phenylalanine, threonine, tryptophan, and valine as nutritionally essential AAs to maintain normal physiological functions of cells and tissues [16,17]. These EAAs are most critical to provide in the diet because a deficiency of any one can limit protein synthesis, which is often manifested as reduced weight gain as well as other specific deficiency signs.

Another group of AAs commonly referred to as dispensable or non-essential amino acids (NEAAs) includes alanine, asparagine, aspartate, cysteine, glutamate, glutamine, glycine, proline, serine, and tyrosine. Those NEAAs traditionally have been classified as such because they can be synthesized in the body from precursor biochemicals. They also may be found in dietary protein and used for synthesizing tissue proteins as well as participating in various metabolic pathways. Emerging evidence has indicated that dietary supplementation of NEAAs also may have beneficial effects on the growth performance [18,19], reproduction [20], health [21,22], and flesh quality [23] of Nile tilapia that are fed plant-based diets. Some AAs may be classified as conditionally essential amino acids (CEAAs) because their rates of use may exceed their rates of synthesis under certain physiological conditions. AAs in this category include glutamate, glutamine, glycine, proline, and hydroxyproline, as well as the sulfonic acid taurine. The CEAA term also has been applied where the reduction or elimination of certain protein feedstuffs in the diet which are rich in certain AAs requires the supplementation of such AAs to prevent growth reduction or other metabolic impairment. The participation of AAs in various metabolic processes beyond protein synthesis such as cell signaling, gene expression, and metabolic regulation has led to the term “functional” AAs, which can include EAAs, CEAAs, and NEAAs [24,25]. The relevance of the concept of functional AAs to aquatic animal nutrition has been established [26] and is beginning to receive heightened attention.

Notably, many relevant outcomes on AA requirements of tilapia have been published since the last National Research Council (NRC) edition in 2011 [27]. However, the summary of AA requirements of tilapia has not been updated for more than a decade. Variables such as fish strain and size, feed composition and processing technology, as well as feed management and statistical methods, may influence experimental estimates of AA requirements of Nile tilapia and may explain some of the inconsistencies in AA recommendations throughout the literature. Importantly, some of these new data provide insights for optimal production efficiency, welfare, health, and flesh quality responses. Thus, this review comprehensively consolidates the relevant data on dietary AA requirements, identifies the shortfalls, and recommends directions for future research in tilapia nutrition.

## 2. Methodology

The studies selected in this review were completed since 1988, and the impacts of single AA additions on the growth performance of Nile tilapia were determined in most of them, although statistical models using the deletion method were applied to estimate AA requirements in the remaining studies. Where data permit, the impacts of dietary AA recommendations based on growth performance, reproduction, and health responses are tabulated. Also, all data are expressed on a dry-matter basis. Generally, growth rate, feed conversion ratio, and protein retention efficiency were identified as the most appropriate parameters for deriving AA recommendations, and their simple mean (±standard deviation) recommendations were used to summarize the AA recommendations reported during the past 34 years. While it is well-established that feedstuffs contain more total than digestible amino acids, few studies have reported dietary amino acid requirements that consider amino acid digestibility. Therefore, in this review, we focused on data based on diets’ total amino acid content. Furthermore, relevant Nile tilapia production stage ranges for the recommendations are included as footnotes. Standard deviations of the mean AA values are recorded to illustrate the variations caused by units in expressing requirements (g/kg diet or % crude protein). These mean values are somewhat problematic because AA requirements are expressed on a total basis, and variations in respective fish size ranges, feed ingredients and diet composition, response variable, and statistical methods for estimating AA requirements are the main influential factors on these outcomes. Recent studies have used various estimation methods to arrive at AA recommendations for Nile tilapia, including linear, quadratic, and polynomial broken-line regressions. An alternative method proposed to determine the optimal AA ratio for pigs [28] and applied for other species such as Atlantic salmon, *Salmo salar* [29], and Nile tilapia [12] is the deletion method. This method involves monitoring the nitrogen balance as AAs are reduced from the diet and assuming that reducing a non-limiting AA does not affect nitrogen retention [30]. This approach differs from the conventional procedure by allowing for the determination of the requirements for all essential AAs in one set of experiments [30] and is well-accepted as an efficient and rapid tool to estimate the ideal AA profile in Nile tilapia [10,11]. Additionally, there are some studies in which more than one assessment method was applied to compare the impacts of different methods on estimated AA requirements. Broken-line models generated 27% of tabulated recommendations, while both linear and broken-line assessments generated 9%. Additionally, polynomial regressions generated 50%, while the deletion method generated 14% of the recommendations. Finally, we established recommendations for the dietary essential AA profile (plus cystine and tyrosine) for each Nile tilapia production stage relative to dietary protein, as these values have not yet been established for all AAs and production stage ranges. For this, we considered previously established dietary protein requirement values in the literature for each production stage.

## 3. Results

### 3.1. Amino Acid Composition of Tilapia Tissues

The AA composition of eggs and whole bodies of Nile tilapia at different production stages are shown in Table 1. Lysine was the dominant essential AA, while glutamic acid was the major non-essential AA. However, there is little information in the literature about the impact of dietary AA supplementation on egg and tissue AA profiles.

Previous research identified that the AA composition of eggs varies with dietary protein [31]. Conversely, early studies have reported that dietary AA supplementation does not alter the whole-body AA composition of nursery, pre-growout, and growout Nile tilapia [35,36,37].

### 3.2. Dietary Amino Acid Recommendations for Nile Tilapia

The AA requirements recommended for some Nile tilapia production stages are presented in Table 2. Of note, there are tangible differences across the literature, and the genesis of these inconsistencies could include genetic differences between strains, diet composition, fish management, rearing conditions, and the statistical method by which AA requirements were assessed. The AA requirements of Nile tilapia were published by the NRC (2011) and are considered one of the main references for AA recommendations. However, genetic improvements and higher performance objectives set by the modern tilapia industry have motivated researchers to review the NRC (2011) recommendations over the past decade. Notably, a number of recent studies have been published, although different experimental methodologies have been used. Several experiments were conducted in which the requirement of one single or multiple AAs was determined, as shown in Table 2.

Based on data displayed in Table 1, the dietary recommendation of each essential AA (plus cystine and tyrosine) was computed relative to the dietary crude protein content and shown in Table 3. For this, we considered the dietary protein content for each fish production stage based on previous values established for Nile tilapia.

Previous evidence shows that the culture system influences the dietary protein requirement of Nile tilapia broodstock. Thus, based on the percentage of each AA relative to dietary protein, the dietary AA recommendations for Nile tilapia broodstock are displayed in Table 4.

The concept of ideal protein for domestic animals was first proposed by Mitchell over 60 years ago [63] and remains relevant in poultry and pig nutrition [64]. The ideal protein concept refers to dietary protein with an AA profile that exactly meets an animal’s requirements [65]. According to this concept, dietary protein should have an AA profile that exactly meets the animal’s requirements, as shown in Table 5. Lysine is used as a reference AA to express the requirements for other AAs, which simplifies diet formulation, as solid requirement data for most AAs are not readily available [66]. Additionally, the concept of ideal AA ratios was introduced for Nile tilapia in 1994 to aid in the development of cost-effective feed formulations [67,68].

Of note, the ideal protein concept also has been applied to reduce dietary protein [69] and optimize fishmeal-free diets [70] for Nile tilapia. Importantly, further studies are required to continuously update ideal AA profiles considering the ideal protein concept in Nile tilapia diets.

### 3.3. Importance of Amino Acids in Nile Tilapia

#### 3.3.1. Lysine

Lysine is the first limiting essential AA in typical corn, wheat, and cereal coproduct–based diets for Nile, and supplementation of feed-grade lysine has been largely adopted in practical and experimental diets for Nile tilapia. Lysine’s primary metabolic role in protein synthesis also is the basis for it to be the reference AA in computing ideal AA ratios. Interestingly, an early study demonstrated that the effectiveness of using intact lysine from high-lysine corn protein concentrate was not significantly different from that of crystalline lysine in Nile tilapia [71]. It is noteworthy that accurate estimations of dietary lysine requirements are critical because recommendations for the balance of AAs based on the ideal protein concept are expressed as ratios to lysine (Table 5). In this sense, dietary lysine is generally considered the first limiting essential AA. As shown in Table 3, there is a relatively low variation in dietary lysine recommendations for body weight gain of fish. On the basis of body weight gain, the simple means of total [45] and digestible lysine [72] intakes are 14 and 12 mg/g body weight gain, respectively. Previous work indicated that lysine utilization efficiency remained relatively high (+63%) in nursery and pre-growout Nile tilapia and decreased (48%) in growout Nile tilapia [73]. In this sense, the dietary lysine requirements for the maintenance of nursery, pre-growout, and growout Nile tilapia have been established to be 2.7, 45.1, and 56.3 mg lysine/kg^0.8^ body, respectively [74]. Previous work determined that growout Nile tilapia required an intake of 23 mg of lysine to deposit 1 g of body weight gain [35]. Moreover, lysine has been shown to improve body weight, feed efficiency, and fillet yield in Nile tilapia [35,75]. Additionally, high quantitative lysine requirements have been described for Nile tilapia reared in saline water [76], with Nile tilapia reared in brackish water (8‰) showing higher lysine requirements than those raised in 0‰ water (23 vs. 21 g/kg diet, respectively) [77].

#### 3.3.2. Sulfur-Containing Amino Acids

Methionine and cysteine (cystine forms from two cysteine residues) are the total sulfur-containing AAs considered in tilapia feeds. Of note, methionine is considered the first limiting AA in Nile tilapia fed cereal-based diets, particularly soybean-meal-rich diets [3,4]. Methionine plays an essential role in cellular metabolism as a methyl donor and acts as the precursor to cysteine [17]. Notably, methionine is one of the most supplemented feed-grade AAs in fish feeds, including those for Nile tilapia [48]. In addition, the total sulfur-containing AA requirement of Nile tilapia is often met by supplementing methionine and considering the replacement values of cysteine and methionine in tandem. Previous research identified that cystine could spare up to 49% of the methionine requirement on a molar sulfur basis in diets for Nile tilapia [48]. Moreover, reports on quantitative methionine requirements for body weight gain are very similar when expressed as individual methionine contents (CV of 13%) compared to methionine-plus-cystine contents (CV of 12%). These suggest that total sulfur-containing AAs may be used to express the dietary requirement, along with considering the minimum level of dietary methionine. The optimal methionine-plus-cystine requirement is well established in the literature, averaging 9.7 g/kg diet (3.5% of crude protein), as displayed in Table 2. A previous study established methionine utilization efficiencies of 0.76 and 0.55 and determined the methionine maintenance requirements of 3.12 and 16.5 mg methionine/kg^0.7^ body for pre-growout and growout Nile tilapia, respectively [78]. Importantly, a previous work determined that growout Nile tilapia required an intake of ~15.2 mg of methionine to deposit 1 g of body weight gain [34].

#### 3.3.3. Threonine

Threonine is a potential limiting AA in conventional corn-, wheat-, soybean-, and coproducts-based diets fed to Nile tilapia [6,7]. Feed-grade threonine is commercially available and may be included in diets for Nile tilapia. Threonine is an important AA because it is prominent in intestinal mucin secretion and in the production of antibodies [79], as well as influencing digestive and absorptive capacities and antioxidant status in the intestine [80]. However, the underlying mechanisms of action of threonine on mucin secretion and intestine health status in Nile tilapia are not fully understood, although many studies have reported the positive impacts of adequate and excess provision of dietary threonine on Nile tilapia performance. Noteworthily, the optimal threonine requirement is well established in the literature and averages 12 g/kg diet (4.3% crude protein), as shown in Table 2. An early study identified the dietary threonine requirement as lower for body weight gain (10.5 g/kg diet; 3.6% crude protein) compared to fillet yield (11.5 g/kg diet; 4% crude protein) in growout Nile tilapia [34]. Additionally, these authors also established that growout Nile tilapia require an intake of 15.9 mg of threonine to deposit 1 g of body weight.

#### 3.3.4. Tryptophan

Tryptophan is considered a potential limiting AA in conventional plant-based diets, particularly in Nile tilapia fed corn and its coproducts [6]. Feed-grade tryptophan is also commercially available for inclusion in Nile tilapia aquafeeds. In addition to protein synthesis, tryptophan plays an important role in producing several metabolites, mainly the neurotransmitter/neuromodulator serotonin and the hormone melatonin in teleost fish [81]. Thus, as a precursor of serotonin, dietary tryptophan has been linked to various behavioral patterns [82]. Consistently, adequate tryptophan supplementation leads to reduced aggressive behavior and stress in Nile tilapia [83], resulting in positive effects on growth, feed efficiency [54], and survival [84]. In this review, the tryptophan requirement averages 3.1 g/kg diet (1% crude protein), as shown in Table 2. Previous work established that pre-growout Nile tilapia require an intake of ~3 mg of tryptophan to deposit 1 g of body weight [52].

#### 3.3.5. Branched-Chain Amino Acids

Isoleucine, leucine, and valine are branched-chain AAs that attract less attention, as their requirements are usually met by typical protein feedstuffs in conventional Nile tilapia diets. In addition, it is important to elaborate diets with a balanced profile of branched-chain AAs because interactions between leucine and valine exist, and imbalances have been reported to depress performance [85]. In this sense, it is possible that branched-chain AA interactions may have influenced the tabulated requirement values. Variations in branched-chain AA recommendations are summarized in Table 2. Growing evidence has suggested the optimal branched-chain AAs ratio (Ile:Leu:Val) to be 1:1.3:0.9 in diets for pre-growout Nile tilapia [13]. However, very few studies have been designed to investigate the interactive effects of branched-chain AAs in Nile tilapia. Feed-grade isoleucine, leucine, and isoleucine have yet to become economically feasible. In addition to protein synthesis, an adequate supply of leucine and valine is important for maintaining immune responses [44,55]. Recently, a study reported that leucine and valine improved intestinal function, enhanced digestive and absorptive capacities, and positively regulated glucose and fatty acid metabolism in the liver, thereby improving the growth performance of Nile tilapia [85]. Dietary isoleucine, leucine and valine requirements average 10.5 g/kg diet (3.8% crude protein), 11.8 g/kg diet (4.2% crude protein), and 10.1 g/kg diet (3.6% crude protein), respectively, as described in Table 2. Previous studies reported that Nile tilapia require intakes of ~15 [13], ~18 [44], and 19 mg [55] of isoleucine, leucine, and valine, respectively, to deposit 1 g of body weight.

#### 3.3.6. Arginine

Arginine has received little consideration in Nile tilapia because its requirement is usually met by typical protein feedstuffs in conventional Nile tilapia aquafeeds. It is noteworthy that the optimal arginine requirement is well established in the literature and averages 14.8 g/kg diet (5.1% crude protein), as shown in Table 2. Early works showed that Nile tilapia required an intake of 20–27 mg of arginine to deposit 1 g of body weight [39,41]. Additionally, a previous study found that arginine at 16.7 g/kg diet stimulated the mRNA expression of myogenic regulatory factors (MRFs), growth hormone (GH), insulin-like growth factors (IGFs) [41], and consequently hypertrophic muscle processes, supporting enhanced growth of Nile tilapia [40]. Also of note, a previous study found that Nile tilapia fed 23.9 g arginine/kg diet exhibited higher immune responses and survival when challenged by Streptococcus agalactiae [86]. Furthermore, emerging evidence suggests that arginine positively changes intestinal microbiota, activates intestinal fatty acid oxidation, and alleviates triglyceride accumulation in intestinal tissue and intracellular cells of Nile tilapia [87]. Additionally, the beneficial effect of 29 g arginine/kg diet on liver health was reported in Nile tilapia reared under high-density (500 fish/m^3^; 63 ± 20 g body weight) conditions in floating net cages [88]. On the contrary, the above authors also reported that a higher level of arginine (41 g/kg diet) promoted the incidence of liver necrosis, further supporting the observation that excess arginine may lead to increased plasma ammonia concentration, decreasing the excretion efficiency of this metabolite, as described in Jian carp, *Cyprinus carpio* var. Jian [89]. However, more extensive research is necessary to investigate the effects of excess arginine on the nitrogen excretion of Nile tilapia.

#### 3.3.7. Histidine

Histidine is considered a marginally limiting AA in typical plant-based protein feedstuffs used in Nile tilapia diets [6,7,90]. Feed-grade histidine is generally not commercially available for use in Nile tilapia aquafeeds. As shown in Table 2, dietary histidine requirements average 7.3 g/kg diet (3.4% crude protein), and previous studies reported that Nile tilapia require an intake of ~9 mg of histidine to deposit 1 g of body weight. An early work reported the positive effects of adequate histidine supplementation on muscle growth by hypertrophy and hyperplasia in pre-growout Nile tilapia [43]. Another study established that histidine also increased mRNA levels of muscle growth-related genes, myoblast determination protein (MyoD), and myogenin, as well as protein synthesis of growout Nile tilapia [42]. Furthermore, emerging evidence has identified the antioxidant capacity of histidine to improve flesh quality attributes in grass carp [91], while another study reported influences on fillet quality of growout Nile tilapia [75]. However, few studies have been conducted to evaluate the health status of Nile tilapia in response to dietary histidine supplementation. This approach is of major importance in applying the precision nutrition concept in Nile tilapia operations.

#### 3.3.8. Total Aromatic Amino Acids

The academic community has not extensively explored the dietary total aromatic AA (phenylalanine plus tyrosine) requirements of Nile tilapia, possibly because both AAs are not considered marginally limiting in feed ingredients typically used for tilapia. Therefore, feed-grade phenylalanine and tyrosine are not commercially available for diet supplementation. However, it is possible that in low-protein diets, phenylalanine may be a limiting AA for Nile tilapia. Importantly, the total aromatic AA requirements of Nile tilapia can be met by supplementing phenylalanine and also considering phenylalanine and tyrosine in tandem because tyrosine can spare some of the dietary phenylalanine otherwise required for tyrosine synthesis [92]. In this regard, the total aromatic AA requirement is influenced by dietary tyrosine levels. Recent work determined the tyrosine replacement value for phenylalanine on a molar basis to be 37% in Nile tilapia [50]. Another recent study confirmed that adequate phenylalanine supplementation influenced growth rate in nursery Nile tilapia [38]. As displayed in Table 2, the optimal phenylalanine-plus-tyrosine requirement averaged 18.3 g/kg diet (5.9% crude protein). A recent study reported that Nile tilapia require an intake of ~35 mg of phenylalanine plus tyrosine to deposit 1 g of body weight [50].

#### 3.3.9. Non-Essential Amino Acids

Growing evidence suggests that non-essential AAs are closely related to the growth performance, health, and flesh quality of Nile tilapia. Non-essential AAs assume more important roles in fish fed plant-rich diets because of their more limited presence [3,4,6,7,90,93]. Additionally, anti-nutritional factors in vegetable ingredients may have adverse effects on AA digestion and absorption and also impair fish health [94]. Emerging evidence has identified that glutamine plays crucial roles in growth and intestinal function [95] and enhances leucocyte function in Nile tilapia [22]. In addition, early studies identified that supplementation of glycine could enhance the antioxidant ability of Nile tilapia [96] and has beneficial effects on growth [19,21]. Previous research reported a positive association between dietary taurine intake and lipid digestion/absorption with carbohydrate and AA metabolism that promoted enhanced growth performance of Nile tilapia [97]. Along with these beneficial effects, taurine plays an antioxidant role [98] which may have positive effects on flesh quality attributes [23] as well as on reproductive performance [20]. Of note, it has been well established that carnitine plays a central role in regulating the lipid β-oxidation of long-chain fatty acids to produce energy in fish species such as zebrafish, *Danio rerio* [99], and this may explain the decreased mesenteric and fillet fat accumulation in growout hybrid tilapia [100]. In addition, carnitine supplementation was found to improve antioxidant functionality in flesh quality attributes [100] and ameliorate or prevent induced liver, intestine, and gill histopathological lesions in Nile tilapia [101].

## 4. Conclusions and Implications

The environmental impact of fish farming is becoming a major challenge that could warrant restrictions on tilapia production. Higher nitrogen excretion levels into the environment are an increasing issue, and well-balanced aquafeeds have been identified as a potential solution. Of note, most studies have estimated the requirement of one individual AA at a time without considering the interactive effects of other dietary AAs. Therefore, it is important to design AA requirement assays for Nile tilapia that take AA interactions into consideration. A lack of attention to protein synthesis dynamics in non-protein-bound AA diets is important because they are supplemented at relatively higher levels in plant-based diets. Therefore, it is important to investigate the digestive dynamics of proteins as well as protein synthesis while maintaining good health status and flesh quality. Moreover, it is evident that AA nutrition of broodstock needs more attention in terms of the impact of AAs on reproductive performance responses. Interestingly, some new studies have reported that non-essential AAs improve growth and reproductive performance as well as the health of fish. Worthy of note is the fact that emerging evidence shows that genomic approaches constitute an important tool for better understanding the underlying mechanisms of fish growth, behavior, and health in order to estimate the AA requirements of Nile tilapia. However, it is clear that the AA requirements of Nile tilapia might vary as a consequence of numerous experimental conditions, including fish strain, size, culture system, and the basal diet used. Notably, genetic selection in Nile tilapia has improved growth performance, fillet yield, and feed efficiency. Therefore, dietary formulations should consider increased amino acid requirements. Interestingly, the current survey highlighted low variations in lysine recommendations relative to protein content, considering that lysine is used as the reference AA in applying the ideal protein concept. This review also identified that emerging studies considered this concept to determine multiple AA requirements using the deletion method. More comprehensive data in the literature are needed for AA requirements through growout in tilapia production in addition to confirming previously established values for lysine, methionine, and threonine. This review observed variations in AA recommendations for different commercial production parameters such as weight gain, feed efficiency, and fillet yield. Therefore, it is important to choose the most appropriate parameter or combination of parameters that represents the business model of the tilapia industry. Finally, these considerations indicated that well-balanced AA diets might be useful for improving the economic and ecological sustainability of tilapia farming into the future.

## Figures and Tables

**Table 1 animals-13-00900-t001:** Amino acid profile (g/100 crude protein) of eggs and whole bodies of Nile tilapia.

AA	Egg ^1^	Fry ^2^	Nursery ^3^	Pre-growout ^4^	Growout ^5^	Mean ± SD
EAA						
Arg	4.7	3.7	4.2	5.4	6.3	4.9 ± 1.0
His	1.9	5.2	2.3	2.1	1.5	2.6 ± 1.5
Ile	2.8	3.2	3.9	3.9	4.3	3.6 ± 0.6
Leu	5.9	6.8	6.0	6.5	7.1	6.5 ± 0.5
Lys	5.8	6.0	7.2	6.7	7.4	6.6 ± 0.7
Met	2.9	5.3	2.0	2.3	2.5	3.0 ± 1.3
Phe	2.2	3.5	3.2	3.7	3.5	3.2 ± 0.6
Thr	4.5	3.7	3.3	3.9	4.1	3.9 ± 0.4
Trp	n.d.	n.d.	0.9	0.9	0.8	0.9 ± 0.1
Val	4.4	4.4	4.4	4.6	4.7	4.5 ± 0.1
NEAA						
Ala	8.4	7.1	5.6	6.1	7.0	6.8± 1.1
Asp	8.1	7.0.	8.2	8.2	8.1	8.2 ± 0.1
Cys	n.d.	n.d.	0.9	0.7	0.7	0.8 ± 0.1
Glu	9.4	11.8	12.1	12.0	14.5	12.0 ± 1.8
Gly	4.3	n.d.	6.8	7.3	9.5	7.0 ± 2.1
Pro	7.2	7.2	n.d	n.d	n.d	7.2 ±0.0
Ser	8.8	5.2	3.2	3.6	3.6	4.9 ± 2.3
Tyr	2.7	3.4	2.9	3.0	2.8	3.0 ± 0.3

Abbreviations: EAA, essential amino acid; NEAA, non-essential amino acid; n.d., non-determined; ^1^ Egg composition of Nile tilapia females fed a 350 g/kg crude protein diet [31]; ^2^ Yolk-sac resorbed Nile tilapia fry of 12 mg body weight [32]; ^3^ Body weight of ~1 g [33]; ^4^ Body weight of ~44 g [13]; ^5^ Body weight of ~829 g [34].

**Table 2 animals-13-00900-t002:** Amino acid recommendations for Nile tilapia.

Amino Acid	Fish Stage	Dietary Requirement		Response	P:E	Reference
		g/kg Diet (DM)	% Protein			
Arg	1	11.8	4.2	WG	26.8	[38]
	1	18.2	6.2	WG	n.p.	[39]
	1	13.6	4.9	WG	21.4	[40]
	2	16.7	5.2	WG	18.5	[41]
	2	13.7	4.9	NR	16.1	[11]
	Mean ± SD	14.8 ± 2.6	5.1 ± 0.7		18.7 ± 2.7	
His	1	4.8	1.7	WG	26.8	[38]
	2	4.8	1.8	NR	16.1	[11]
	2	8.2	3.1	WG	15.4	[42]
	3	8.8	2.8	WG	19.4	[43]
	Mean ± SD	6.7 ± 2.2	2.4 ± 0.7		19.4 ± 5.2	
Ile	1	8.7	3.1	WG	26.8	[38]
	2	9.1	3.3	NR	16.1	[11]
	1	13.7	5.0	WG	19.6	[13]
	Mean ± SD	10.5 ± 2.8	3.8 ± 1.0		20.8 ± 5.5	
Leu	1	9.5	0.34	WG	26.8	[38]
	2	13.5	0.48	NR	16.1	[11]
	1	12.5	0.43	WG	n.p.	[44]
	Mean ± SD	11.8 ± 2.1	4.2 ± 007		21.5 ± 7.6	
Lys	1	14.3	5.1	WG	26.8	[38]
	2	16.5	5.9	NR	16.1	[11]
	3	15.1	6.0	WG	19.2	[35]
	2	18.0	5.6	WG	18.5	[45]
	Mean ± SD	16.0 ± 1.6	5.7 ± 0.4		20.2 ± 4.6	
Met	1	7.5	2.7	WG	26.8	[38]
	3	6.8	2.3	WG	20.6	[46]
	1	9.1	3.2	WG	n.p.	[47]
	1	8.1	2.9	WG	19.1	[48]
	Mean ± SD	7.9 ± 1.0	2.8 ± 0.4		22.2 ± 4.1	
Met +Cys	1	9.0	3.2	WG	26.8	[38]
	3	11.2	3.8	WG, FY	20.6	[46]
	1	10.0	3.5	WG	n.p.	[47]
	1	8.5	3.0	WG	19.1	[48]
	Mean ± SD	9.7 ± 1.2	3.4 ± 0.4		22.2 ± 4.1	
Phe	1	10.5	3.8	WG	26.8	[38]
	1 ^1^	8.8	3.0	WG	21.4	[49]
	1	12.1	3.5	WG	20.9	[50]
	Mean ± SD	10.5 ± 1.2	3.4 ± 0.4		23.0 ± 3.3	
Phe + Tyr	1	15.5	5.6	WG	26.8	[38]
	1 ^1^	18.6	6.4	WG		[49]
	1	20.6	5.9	WG		[50]
	Mean ± SD	18.2 ± 2.6	6.0 ± 0.4		23.0 ± 3.3	
Thr	1	10.5	3.8	WG	26.8	[38]
	1	13.3	4.7	WG		[51]
	2	13.5	4.8	NR	16.1	[11]
	3	11.5	4.0	WG, FY	22.7	[34]
	Mean ± SD	12.2 ± 1.4	4.3 ± 0.5		21.9 ± 5.4	
Trp	1	2.8	1.0	WG	26.8	[38]
	2	2.4	0.9	NR	16.1	[11]
	1	3.4	1.1	WG	19.3	[52]
	1	3.8	1.2	WG	18.5	[53]
	1	3.1	1.0	WG	n.p.	[54]
	Mean ± SD	3.1 ± 0.5	1.0 ± 0.1		20.2 ± 4.6	
Val	1	7.8	2.8	WG	26.8	[38]
	2	9.7	3.5	NR	16.1	[11]
	1	12.7	4.5	WG	17.4	[55]
	Mean ± SD	10.1 ± 2.5	3.6 ± 0.9		20.1 ± 5.8	

Abbreviations: SD, standard deviation; DM, dry matter; WG, weight gain; NR, nitrogen retention; FY, fillet yield; P:E, protein-to-energy ratio (g/MJ gross or digestible energy); n.p., non-presented; ^1^ Hybrid tilapia, *Oreochromis niloticus* × *Oreochromis aureus*.

**Table 3 animals-13-00900-t003:** Dietary amino acid recommendation (g/kg diet) for Nile tilapia ^1^.

Amino Acid	Production Stage ^2^		
	Fry	Nursery/Pre-Growout	Growout
	Crude Protein (g/kg Diet) ^3^		
	460	350	320
Arg	23.5	17.9	16.3
His	11.0	8.4	7.7
Ile	17.5	13.3	12.2
Leu	19.3	14.7	13.4
Lys	26.2	20.0	18.2
Met	12.9	9.8	9.0
Met + Cys	15.6	11.9	10.9
Phe	15.6	11.9	10.9
Phe + Tyr	27.6	21.0	19.2
Thr	19.8	15.1	13.8
Trp	0.5	0.4	0.3
Val	16.1	12.3	11.2

^1^ Each amino acid is estimated relative to crude protein (mean value) displayed in Table 2; ^2^ Previously established for Nile tilapia as nursery (1.6 to 30 g of body weight), pre-growout (31 to ≤220 g of body weight), and growout (>220 g of body weight) [56]; ^3^ Mean values previously established for Nile tilapia fry [57,58], nursery and pre-growout [57], and growout Nile tilapia [59].

**Table 4 animals-13-00900-t004:** Dietary amino acid recommendations (g/kg diet) for broodstock Nile tilapia ^1^.

Amino Acid	Crude Protein (g/kg Diet; Dry Matter)		
	350 ^2^	380 ^3^	400 ^4^
Arg	17.9	19.4	20.4
His	8.4	9.1	9.6
Ile	13.3	14.4	15.2
Leu	14.7	16.0	16.8
Lys	20.0	21.7	22.8
Met	9.8	10.6	11.2
Met + Cys	11.9	12.9	13.6
Phe	11.9	12.9	13.6
Phe + Tyr	21.0	22.8	24.0
Thr	15.1	16.3	17.2
Trp	0.4	0.4	0.4
Val	12.3	13.3	14.0

^1^ Each amino acid is estimated relative to crude protein (mean value) displayed in Table 2; ^2^ Broodstock raised in earthen pond [31,60]; ^3^ Broodstock raised in recycling system [61]; ^4^ Broodstock raised in water salinity up to 14‰ [62].

**Table 5 animals-13-00900-t005:** Dietary amino acid profile (% of lysine) based on the ideal protein concept for Nile tilapia.

Amino Acid	Fish Production Stage ^1^			Mean ± SD
	Nursery ^2^	Pre-Growout ^2^	Growout ^2^	
Lysine	100	100	100	10 0 ± 0
Arginine	86	125	81	97 ± 24
Histidine	30	34	34	33 ± 2
Isoleucine	56	57	51	55 ± 3
Leucine	84	96	66	82 ± 15
Methionine	41	64 ^3^	41 ^3^	49 ± 13
Phenylalanine	64	101 ^4^	70 ^4^	78 ± 20
Threonine	103	93	89	95 ± 7
Tryptophan	16	24	23	21 ± 4
Valine	60	76	73	70 ± 9

Abbreviation: SD, standard deviation; ^1^ Previously established for Nile tilapia as nursery (1.6 to 30 g of body weight), pre-growout (31 to ≤220 g of body weight), and growout (>220 g of body weight) [56]; ^2^ Data established for nursery [11], pre-growout [12], and growout [10] Nile tilapia by deletion method; ^3^ Methionine plus cystine; ^4^ Phenylanaline plus tyrosine.

## Data Availability

The data presented in this study are available on request from the corresponding author.

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
