# Peer review of "Amino Acid Requirements for Nile Tilapia: An Update"

_animals, 2023, doi:10.3390/ani13050900_

Round 1
Reviewer 1 Report
The author reviewed and concluded: (1) The data on dietary amino acid (AA) requirements of tilapia of three production stages and under broodstock condition; (2) Dietary AA profile based on the ideal protein concept; (3) Several factors such as strain, size, basal diet composition, and assessment method affect AA evaluation; (4) The importance of different AAs. The author introduced some shortages of the present researches and suggested that more studies should focus on the AA interactions, protein synthesis dynamics, and AA nutrition of broodstock. In addition, it is suggested that ideal protein concept and deletion method technique be used in the evaluation of amino acid requirements of tilapia. The review is well-documented and has reference significance for production and future research on amino acid requirements on tilapia.
Line 129 Add a space between “recommendations” and “for”.
Author Response
We thank the Reviewer for his thoughtful and thorough review and believe his input has been invaluable to make our review more balanced. In this new version, we have tried to improve the manuscript presentation.
Reviewer
Line 129 Add a space between “recommendations” and “for”.
Response:
We apologize for our inaccuracies and thank the Reviewer for pointing this out. We have corrected the sentences in the revised manuscript according to the Reviewer’s comments.
Reviewer 2 Report
The paper is well-written, clear and presented the most recent findings on amino acid nutrition for Nile tilapia, an important aquacultured fish. Results were presented in a summarized and logical way, leading to a comprehensive understanding of the problem addresed by the authors. I really enjoyed the reading and have very few comments to the manuscript.
1) I only missed more information regarding differences on the amino acid requirements according to the fish strains. The authors should have explored this since this effect is not well-explored in aquacultured fish species. A graphic depicting the main differences on amino acid requirement among the most farmed tilapia strains would be great since the authors explored these differences in several topics of this review. Nonetheless, some emphasis to have been made on the digestible/ crude nutrient levels. It is not clear if the outcomes the authors have come after the literature review considered the digestibility of the protein and/or amino acids. Althoguh this might be evident for fish nutritionists, it may mislead some readers when using certain types of protein sources.
2) Additionally, I only found some typos during the evaluation of the manuscript. I recommend the authors revise the word "nursery". I found it incorrectly written in one paragraph and one table heading.
Author Response
1) I only missed more information regarding differences on the amino acid requirements according to the fish strains. The authors should have explored this since this effect is not well-explored in aquacultured fish species. A graphic depicting the main differences on amino acid requirement among the most farmed tilapia strains would be great since the authors explored these differences in several topics of this review. Nonetheless, some emphasis to have been made on the digestible/ crude nutrient levels. It is not clear if the outcomes the authors have come after the literature review considered the digestibility of the protein and/or amino acids. Althoguh this might be evident for fish nutritionists, it may mislead some readers when using certain types of protein sources.
Response: (Reviewer) - I only missed more information regarding differences on the amino acid requirements according to the fish strains. The authors should have explored this since this effect is not well-explored in aquacultured fish species. (Response) - Thank you to the Reviewer for pointing this out. We have included these pieces of information in the revised manuscript. (Reviewer) - A graphic depicting the main differences on amino acid requirement among the most farmed tilapia strains would be great since the authors explored these differences in several topics of this review. (Response) - Thank you for pointing this out. Unfortunately, due to database limitations, it was not possible to produce a robust figure demonstrating differences between tilapia strains, accounting for variations in size, rearing systems, analysis, statistics, and experimental diets used. Nevertheless, we believe that in the near future a figure for some amino acids, such as lysine, will be possible. (Reviewer) - Nonetheless, some emphasis to have been made on the digestible/ crude nutrient levels. It is not clear if the outcomes the authors have come after the literature review considered the digestibility of the protein and/or amino acids. Althoguh this might be evident for fish nutritionists, it may mislead some readers when using certain types of protein sources. (Response) -Thank you to the Reviewer for pointing this out. We have included these pieces of information in the revised manuscript.
Reviewer: 2) Additionally, I only found some typos during the evaluation of the manuscript. I recommend the authors revise the word "nursery". I found it incorrectly written in one paragraph and one table heading.
Response. Thank you to the Reviewer for pointing this out. We have corrected this error thoughtful the revised manuscript.
Reviewer 3 Report
1/ In Tables 3, 4 and 5, units for AA should be clearly stated. It is inferred that there are g/kg (as for Crude Protein), but I think that they should be clearly stated and visible.
2/ Authors should also make a comment on the Energy values of the selected referenced diets : are the used diets (in bibliography) energetically sufficient ? In other words, which is the Protein/Energy ratio of the used diets and is it the optimum and uniform among experiments. The Heat Increment is a major criterion in relation to the dietary Protein Efficiency and Protein Retention values.
Author Response
(Reviewer) 1/ In Tables 3, 4 and 5, units for AA should be clearly stated. It is inferred that there are g/kg (as for Crude Protein), but I think that they should be clearly stated and visible.(Response) - We thank the Reviewer for these suggestions. We have considered this suggestion in the revised manuscript.
2/ Authors should also make a comment on the Energy values of the selected referenced diets : are the used diets (in bibliography) energetically sufficient ? In other words, which is the Protein/Energy ratio of the used diets and is it the optimum and uniform among experiments. The Heat Increment is a major criterion in relation to the dietary Protein Efficiency and Protein Retention values. (Response) - We think this is an excellent suggestion. We have included data of protein:energy ratio in Table 2.
